# Impaired bone strength and bone microstructure in a novel early-onset osteoporotic rat model with a clinically relevant *PLS3* mutation

**Jing Hu[1], Bingna Zhou[1], Xiaoyun Lin[1], Qian Zhang[1], Feifei Guan[2], Lei Sun[1], Jiayi Liu[1], Ou Wang[1], Yan Jiang[1], Wei-bo Xia[1], Xiaoping Xing[1], Mei Li[1]***

[1]Department of Endocrinology, Key Laboratory of Endocrinology, National Health and Family Planning Commission, Peking Union Medical College Hospital, Chinese Academy of Medical Sciences and Peking Union Medical College, Beijing, China; [2]Key Laboratory of Human Disease Comparative Medicine, NHFPC, Institute of Laboratory Animal Science,Chinese Academy of Medical Sciences & Comparative Medical Center, Peking Union Medical College, Beijing, China

**Abstract** Plastin 3 (PLS3), a protein involved in formation of filamentous actin (F-actin) bundles, is important in human bone health. Recent studies identify *PLS3* as a novel bone regulator and *PLS3* mutations can lead to a rare monogenic early-onset osteoporosis. However, the mechanism of *PLS3* mutation leading to osteoporosis is unknown, and its effective treatment strategies have not been established. Here, we have constructed a novel rat model with clinically relevant hemizygous E10-16del mutation in *PLS3* (*PLS3*^(E10-16del/0)) that recapitulates the osteoporotic phenotypes with obviously thinner cortical thickness, significant decreases in yield load, maximum load, and breaking load of femora at 3, 6, 9 months old compared to wild-type rats. Histomorphometric analysis indicates a significantly lower mineral apposition rate in *PLS3*^(E10-16del/0) rats. Treatment with alendronate (1.0 µg/kg/day) or teriparatide (40 µg/kg five times weekly) for 8 weeks significantly improves bone mass and bone microarchitecture, and bone strength is significantly increased after teriparatide treatment ($p < 0.05$). Thus, our results indicate that *PLS3* plays an important role in the regulation of bone microstructure and bone strength, and we provide a novel animal model for the study of X-linked early-onset osteoporosis. Alendronate and teriparatide treatment could be a potential treatment for early-onset osteoporosis induced by *PLS3* mutation.

*For correspondence: limeilzh@sina.com

Competing interest: The authors declare that no competing interests exist.

## Editor's evaluation

The findings are significant with regard to the clinical treatment of early onset osteoporosis due to PLS3 mutations. The evidence which supports the conclusions of the manuscript is strong. This manuscript will be of notable relevance to the field of metabolic bone diseases.

## Introduction

Osteoporosis is the most common metabolic bone disorder and is characterized by low bone mineral density (BMD), bone microarchitecture deterioration, and increased predisposition for bone fractures. Osteoporosis is usually regarded as an aging-related disease, however, idiopathic osteoporosis has also emerged in children and adolescents, which refers to significantly lower-than-expected bone mass manifesting in childhood with no identifiable etiology (*Mäkitie and Zillikens, 2022*). Recently,

genetic variations have been found to be closely associated with diversities of BMD, fracture risk, and effects of anti-osteoporotic treatment (*Ralston and Uitterlinden, 2010*). A series of studies identify multiple heritable genetic variants that lead to early-onset osteoporosis (EOOP).

In 2013, a new kind of severe EOOP was first described, which was caused by mutations in *PLS3* (*van Dijk et al., 2013*). *PLS3* (OMIM 300131) has 16 exons and spans approximately 90 kb, which is located on Xq23 and encodes a highly conserved protein plastin 3 (PLS3). PLS3 belongs to a family of actin-binding proteins, which is ubiquitously expressed in solid tissues and involved in the binding and bundling of actin filaments in the cytoskeleton, thus partaking in various cellular functions, such as cell migration and adhesion (*Wolff et al., 2021*). In bone, PLS3 is proposed to regulate cytoskel-etal actin bundling, osteocyte function and their mechanosensory apparatus, osteoclast function, and bone matrix mineralization (*Pathak et al., 2020*; *Wolff et al., 2021*). A series of pedigree studies demonstrated that variants in *PLS3* led to significantly reduced BMD and early-onset recurrent fragility fractures (*Costantini et al., 2018*; *Fahiminiya et al., 2014*; *Hu et al., 2020*; *Kämpe et al., 2017*; *Kannu et al., 2017*; *Laine et al., 2015*; *Lv et al., 2017*; *van Dijk et al., 2013*; *Wang et al., 2020*). Till now, 29 different mutations in *PLS3* have been reported to induce EOOP (*Brlek et al., 2021*; *Cohen et al., 2022*; *Wolff et al., 2021*). Due to its X-chromosomal inheritance, *PLS3*-induced osteoporosis has more severe effects on males than females, although heterozygous carrier females also suffer from EOOP. However, the exact function of PLS3 in bone is still unknown.

So far, the animal models carrying patient-derived *PLS3* mutations have not been generated, which are valuable to unveil the pathogenesis of this ultra-rare X-linked osteoporosis induced by *PLS3* mutations. A previously reported *PLS3* knock-out (KO) murine model displayed a significant decrease in cortical thickness (Ct.Th) with normal or decreased trabecular number. This model harbored a complete deletion of *PLS3* gene that was different from mutations identified in patients (*Neugebauer et al., 2018*; *Yorgan et al., 2020*). Moreover, effective treatment strategies have not been established in EOOP related to *PLS3* mutations. Few patients with *PLS3*-related EOOP received bisphosphonates or teriparatide (TPTD) treatment, while the efficacy was variable (*Fratzl-Zelman et al., 2021*; *Hu et al., 2020*; *Lv et al., 2017*; *Välimäki et al., 2017*; *van Dijk et al., 2013*). To explore the potential pathogenesis and treatment strategies of *PLS3*-related EOOP, we independently constructed a novel rat model with ubiquitous deletion of the exon 10–16 of *PLS3* (*PLS3*$^{E10\text{-}16del/0}$), which recapitulated a patient-specific mutation of exon 10–16 deletion in *PLS3* (*Lv et al., 2017*). We observed that *PLS3*$^{E10\text{-}16del/0}$ rats, similar to the respective patient, displayed impaired bone strength due to thinner cortical bone, which could be improved by treatment with alendronate (ALN) and TPTD. Compared to the previously reported mice model, the novel rat model had a milder bone phenotype possibly due to the presence of truncated PLS3 variants.

## Results

### The rats with hemizygous E10-16del mutation were successfully generated

A large fragment deletion of exon 10–16 in *PLS3* was introduced into the genome of rats, and a 9626 bp deletion from 84,172 to 93,797 bp (NC_051356.1) was confirmed by genotyping and Sanger sequencing (*Figure 1A and B*). PLS3 antibody used in this study targeted the KO region of PLS3 protein (deleted amino acid region: 331–630). Therefore, using western blotting of equivalent amounts of total proteins from wild-type (WT) and *PLS3*$^{E10\text{-}16del/0}$ rats, we observed a band of appropriate size for PLS3 (~70 kDa) in membranes of WT rats, which was lacking in membranes of *PLS3*$^{E10\text{-}16del/0}$ rats (*Figure 1C*). Immunohistochemical staining of femoral sections also unraveled that PLS3 was present in osteocytes, osteoblasts, and osteoclasts in cortical and trabecular bone of WT rats, while PLS3 was not seen in bone cells of the *PLS3*$^{E10\text{-}16del/0}$ rats (*Figure 1E*). Moreover, quantitative polymerase chain reaction (qPCR) results also indicated that the expression level of *PLS3 E10-16* was extremely low in *PLS3*$^{E10\text{-}16del/0}$ rats. However, we found a similar expression level of *PLS3* E1-9 between *PLS3*$^{E10\text{-}16del/0}$ and WT rats, which indicated a possible presence of truncated PLS3 variants (*Figure 1D*). Together, these results indicated that rats with hemizygous E10-16 deletion in *PLS3* were successfully built.

Rats carrying mutant allele were born close to the expected Mendelian ratio (*PLS3*$^{E10\text{-}16del/+}$: 28.6%, *PLS3*$^{E10\text{-}16del/0}$: 28.5%, WT: 42.4%). The *PLS3*$^{E10\text{-}16del/0}$ rats showed similar movement, locomotor activi-ties, and longitudinal growth to WT rats (*Figure 1—figure supplement 1*). Moreover, the life span of

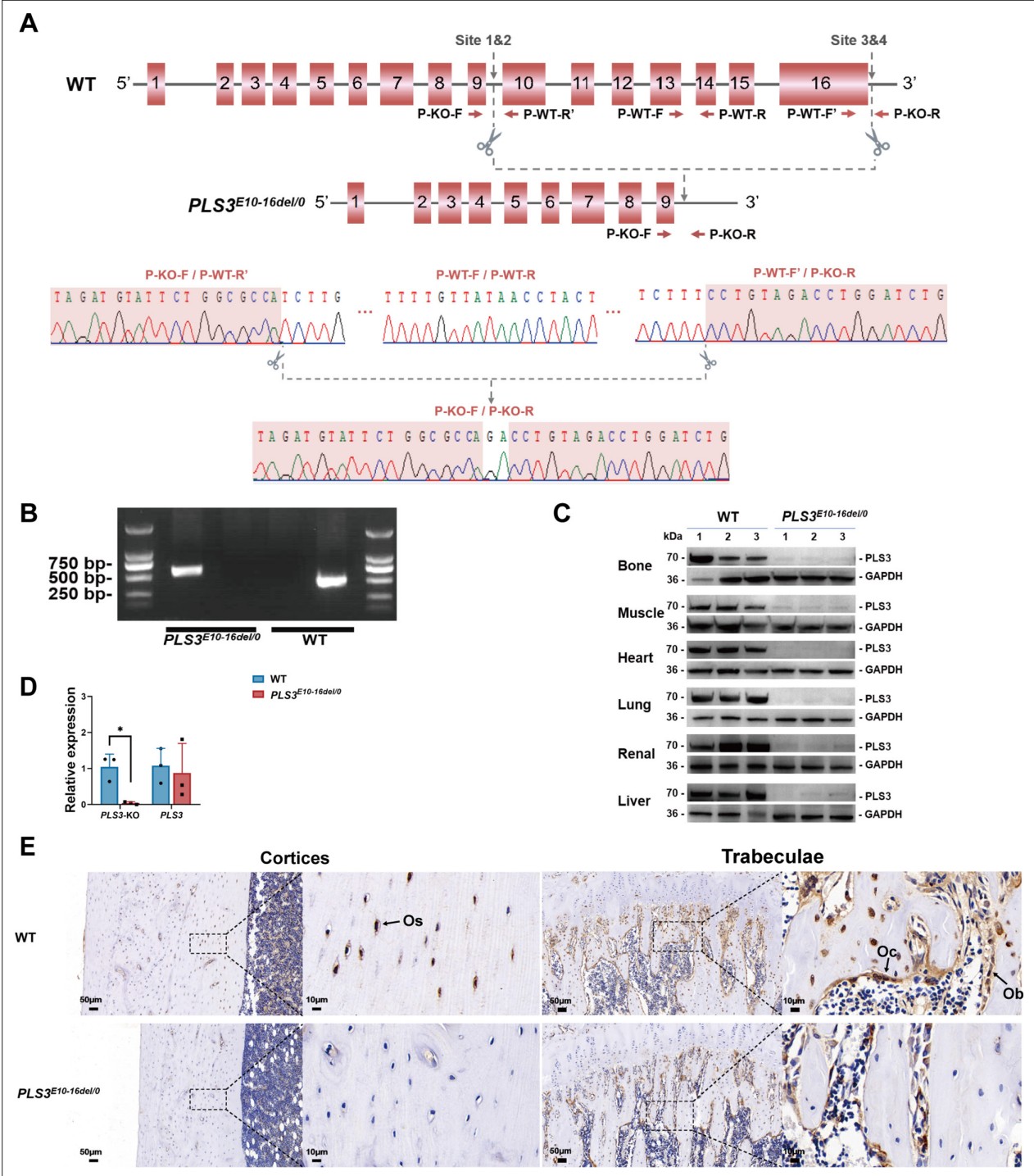

**Figure 1.** Rats with *PLS3* E10-16del mutation built by CRISPR/Cas9 and confirmation. (**A**) Schematic diagram of targeted *PLS3* gene deletion using CRISPR/Cas9 and Sanger sequencing analysis. Four specific CRISPR target sites near the genomic region of exon 10–16 were designed. CRISPR/Cas9 systems specifically cleaved the target sites. Sanger sequence confirmed the genomic deletion of exon 10–16 (No. 84172–93797 bp) and two base pair insertions at the target site due to non-homologous end joining after the DNA cleavage. (**B**) Polymerase chain reaction (PCR) genotyping of *PLS3*<sup>E10-16del/0</sup> rats. DNA from tail snips was subjected to PCR using P-KO-F/P-KO-R and P-WT-F/P-WT-R primers for mutant and WT allele, respectively. Amplification of mutant samples resulted in one copy of the upper 554 bp fragment, while WT samples yield one copy of the lower 450 bp fragment. (**C**) *PLS3* gene deletion confirmed by western blot. PLS3 protein expression was absent in various tissues of the *PLS3*<sup>E10-16del/0</sup> rats compared to age-matched WT rats, indicating the whole-body deletion of the *PLS3* gene. (**D**) Quantitative PCR (qPCR) confirmation of *PLS3* E10-16 deletion. The results of *PLS3*-KO indicated the expression level of *PLS3* E10-16 (knockout region). The results of *PLS3* indicated the expression level of *PLS3* E1-9 (uneditable region). The expression level of *PLS3 E10-16* was extremely low in *PLS3*<sup>E10-16del/0</sup> rats while the expression level of *PLS3* exon 1–9 was similar to WT rats. Data were

*Figure 1 continued on next page*

*Figure 1 continued*

pooled from three independent experiments and were presented as mean ± SEM. Data were analyzed using unpaired two-tailed Student t test. *p＜0.05 vs WT groups. (**E**) Representative image of PLS3 immunohistochemistry in the femoral sections. Os, osteocytes; Ob, osteoblasts; Oc, osteoclasts. Immunohistochemical staining images revealed that PLS3 was present in osteocytes, osteoblasts, and osteoclasts in the cortical (left) and trabecular (right) bone of WT rats. Loss of staining in osteocytes, osteoblasts, and osteoclasts in the *PLS3^E10-16del/0* rats confirmed the successful deletion of the *PLS3* gene in bone. WT: wild-type.

The online version of this article includes the following source data and figure supplement(s) for figure 1:

**Source data 1.** Sanger sequencing confirmation of PLS3 E10-16 knockout.

**Source data 2.** Western blotting analysis of PLS3 protein expression.

**Source data 3.** Original data of quantitative polymerase chain reaction (qPCR) results.

**Figure supplement 1.** Growth curve of *PLS3^E10-16del/0* and WT rats (n = 5 per group).

WT and *PLS3^E10-16del/0* rats was similar and no rats died during the experimental period. There were also no statistical differences in body weight change between WT and *PLS3^E10-16del/0* rats during the whole experimental period (*Figure 1—figure supplement 1*).

The biomechanical properties of *PLS3^E10-16del/0* rats were significantly impaired. Compared to age- and gender-matched WT group, *PLS3^E10-16del/0* rats exhibited a decrease by 24.8% in yield load (p＜0.001), 19.4% in maximum load (p<0.01), and 25.3% in breaking load (p<0.05) of femur of 3-month-old *PLS3^E10-16del/0* rats, which continued to 6 and 9 months old of mutant rats (*Figure 2A and B*). Notably, the stiffness (108.18±11.28 vs 501.20±84.97 N/mm, p＜0.001) and breaking load (155.99±39.92 vs 227.75±27.06 N, p＜0.01) of femur of 9-month-old *PLS3^E10-16del/0* rats were pronounced lower than those of WT rats. Significantly reduced maximum load of vertebrae was also found in *PLS3^E10-16del/0* rats (*Figure 2C*). Compared with WT rats, the average maximum load of the fifth lumbar vertebra decreased by 28.5% (p＜0.01) and 43.6% (p＜0.001) at 3 and 6 months old of *PLS3^E10-16del/0* rats. No significant difference was found in the stiffness, work-to-failure, and post-yield displacement between 3-month-old *PLS3^E10-16del/0* rats and WT rats (*Figure 2A*, *Figure 2—figure supplement 1A*).

The microstructure of cortical bone was deteriorated in *PLS3^E10-16del/0* rats. Femoral Ct.Th of 3-, 6-, 9-month-old *PLS3^E10-16del/0* rats were 79.9% (p＜0.001), 86.3% (p＜0.05), 72.6% (p＜0.001) of age-matched WT rats, respectively. However, bone volume/tissue volume (BV/TV), bone surface area/bone volume (BS/BV), trabecular thickness (Tb.Th), trabecular number (Tb.N), and trabecular separation (Tb.Sp) in femur of *PLS3^E10-16del/0* rats were similar to WT rats at all ages (*Figure 2D and E*, *Figure 2—figure supplement 1B*). No significant differences were found in %Tb.Ar, Tb.Th, Tb.N, and Tb.Sp of lumbar vertebrae between the *PLS3^E10-16del/0* rats and WT rats (*Figure 2—figure supplement 1C*).

The mineral apposition rate (MAR) of trabecular bone (Tb.MAR) in lumbar vertebrae was significantly decreased in 6- and 9-month-old *PLS3^E10-16del/0* rats than WT rats (*Figure 2G*), while no statistical changes were detected in MAR of endocortical (Ec.MAR) and periosteal surface (Ps.MAR) of tibial cortex (*Figure 2—figure supplement 1D*). The number of osteocytes, osteoclasts, and osteoblasts in *PLS3^E10-16del/0* rats was similar to WT rats at all ages (*Figure 2—figure supplement 1E*). Also, *PLS3^E10-16del/0* and WT rats had similar Ocn-positive areas at the trabecular and cortical bone of femur (*Figure 2—figure supplement 2A*). Serum levels of total alkaline phosphatase (ALP), β-CTX, and calcium were similar in *PLS3^E10-16del/0* and WT rats (*Figure 2—figure supplement 2C*).

Since no difference was found in bone mass between WT and *PLS3^E10-16del/0* rats, we further investigated the quantity and structure of bone collagen. The expression of *COL1A1* in tibia was also similar between two groups. However, compared to WT rats, the cortical bone was more porous and the collagen fibers of *PLS3^E10-16del/0* rats were relatively disorganized (*Figure 2H*).

No difference was found in the area of adipocytes in the distal marrow per tissue area. However, 3-month-old *PLS3^E10-16del/0* rats had more adipocytes than WT rats of the same age (*Figure 2—figure supplement 2B*). We also investigated the parameters of glucose and lipid metabolism and found similar levels of serum glucose (Glu), triglycerides (TG), cholesterol (TC), low-density lipoprotein (LDL) between *PLS3^E10-16del/0* rats and WT at all ages (*Figure 2—figure supplement 2D*). Besides, *PLS3* mutations were reported to have patent ductus arteriosus (*Qiu et al., 2022*). However, no obvious abnormalities were found in ultrasonic cardiogram and hematoxylin and eosin (H&E) staining about the cardiac structure and function of *PLS3^E10-16del/0* rats.

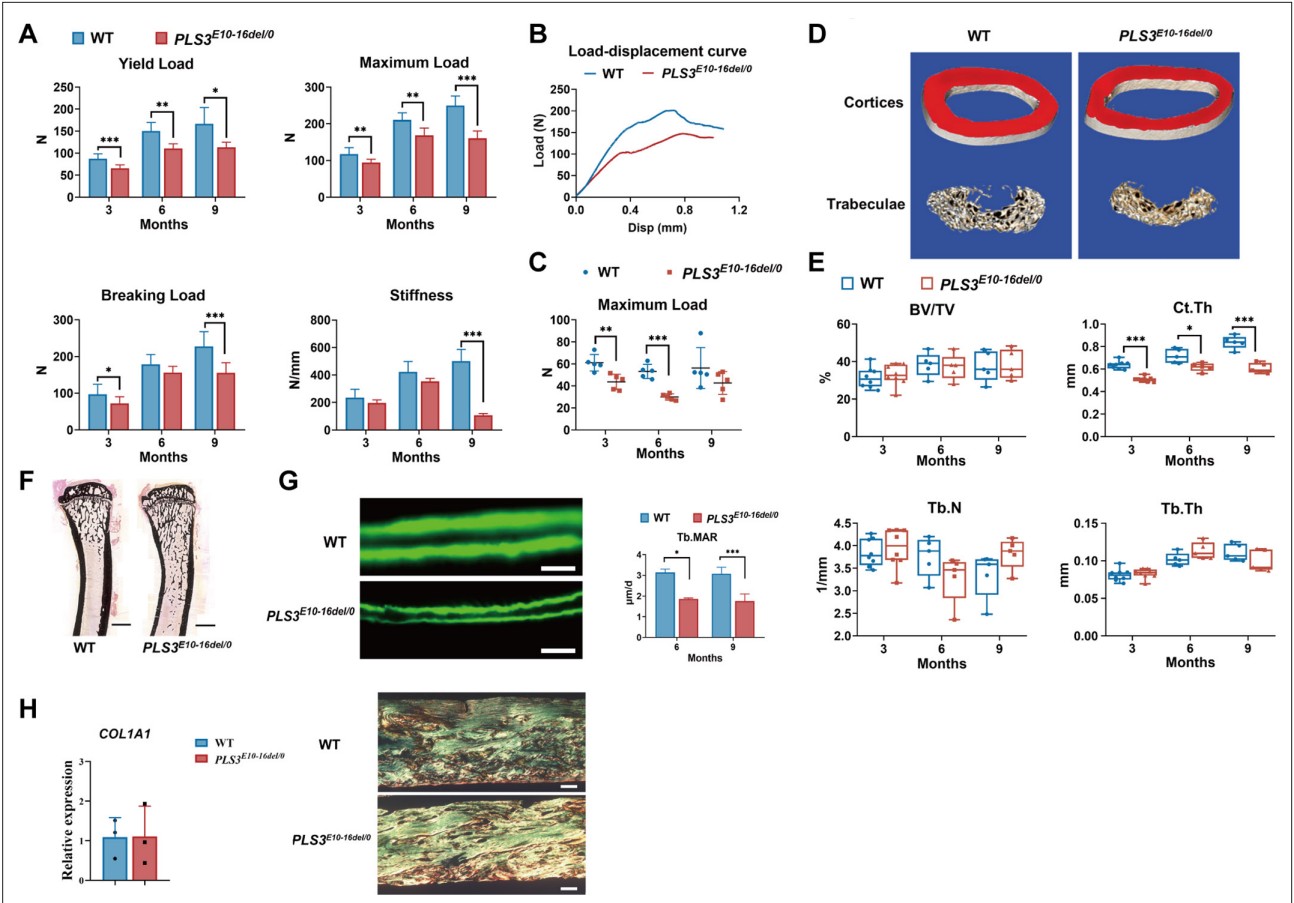

**Figure 2.** Bone strength, bone microstructure, and bone formation activity of *PLS3^E10-16del/0* rats. (**A**) Mechanical three-point bending tests of femora from *PLS3^E10-16del/0* and WT rats (n=5-8 per group). (**B**) Typical load-displacement curves of *PLS3^E10-16del/0* and WT rats. (**C**) Indentation tests of L$_5$ from *PLS3^E10-16del/0* and WT rats (n=5 per group). Data were analyzed using unpaired two-tailed Student t test. *p<0.05, **p <0.01, ***p <0.001 vs WT groups. (**D**) Three-dimensional reconstruction images of femurs from *PLS3^E10-16del/0* and WT rats. (**E**) Micro-CT assessment of the distal femurs from *PLS3^E10-16del/0* and WT rats (n=5-8 per group). BV/TV: bone volume/tissue volume, Ct.Th: cortical thickness, Tb.Th: trabecular thickness, Tb.N: trabecular number. (**F**) Representative von Kossa-stained sections of tibia diaphysis of *PLS3^E10-16del/0* and WT rats. Scale bar = 2000 µm. WT: wild-type. (**G**) Typical images of unstained and uncalcified vertebra of *PLS3^E10-16del/0* rats and comparison of mineral apposition rate (n=4 per group). Scale bar = 10 µm. Data were analyzed using unpaired two-tailed Student t test. *p<0.05 vs WT groups. Tb.MAR: mineral apposition rate of lumbar trabeculae. (**H**) Expression level of *COL1A1* and photomicrographs of picrosirius red-stained sections of cortical bone regions of femur visualized through polarized light microscopy. Scale bar = 100 µm. Data were pooled from three independent experiments and were presented as mean ± SEM.

The online version of this article includes the following source data and figure supplement(s) for figure 2:

**Source data 1.** Original data of the results of three-point binding tests.

**Source data 2.** Original data of the results of indentation testing.

**Source data 3.** Micro-computed tomography (µCT) analysis of the distal femur and histomorphometric evaluation of L4.

**Source data 4.** Original data of histomorphometry.

**Figure supplement 1.** Bone strength and bone morphometry parameters of *PLS3^E10-16del/0* rats.

**Figure supplement 2.** Histological, immunohistochemical, and serum biochemical analysis of *PLS3^E10-16del/0* rats.

## Effects of anti-osteoporotic treatment on *PLS3^E10-16del/0* rats

ALN or TPTD treatment for 8 weeks significantly improved bone microstructure of *PLS3^E10-16del/0* rats. Compared to vehicle (VEH) group, Tb.N and BV/TV values were increased by 38.4% and 38.7% in ALN group and by 35.9% and 29.3% in TPTD group, while Tb.Sp was decreased by 49.3% and 40.1% in ALN and TPTD group, respectively (all p<0.05), and cancellous BMD of femur increased by 43.0% in ALN group (p<0.01) and 33.3% in TPTD group (p<0.01). ALN or TPTD treatment significantly increase 7.4% and 4.8% of Ct.Th of *PLS3^E10-16del/0* rats (all p<0.05 vs VEH group) (*Figure 3A and B*). A

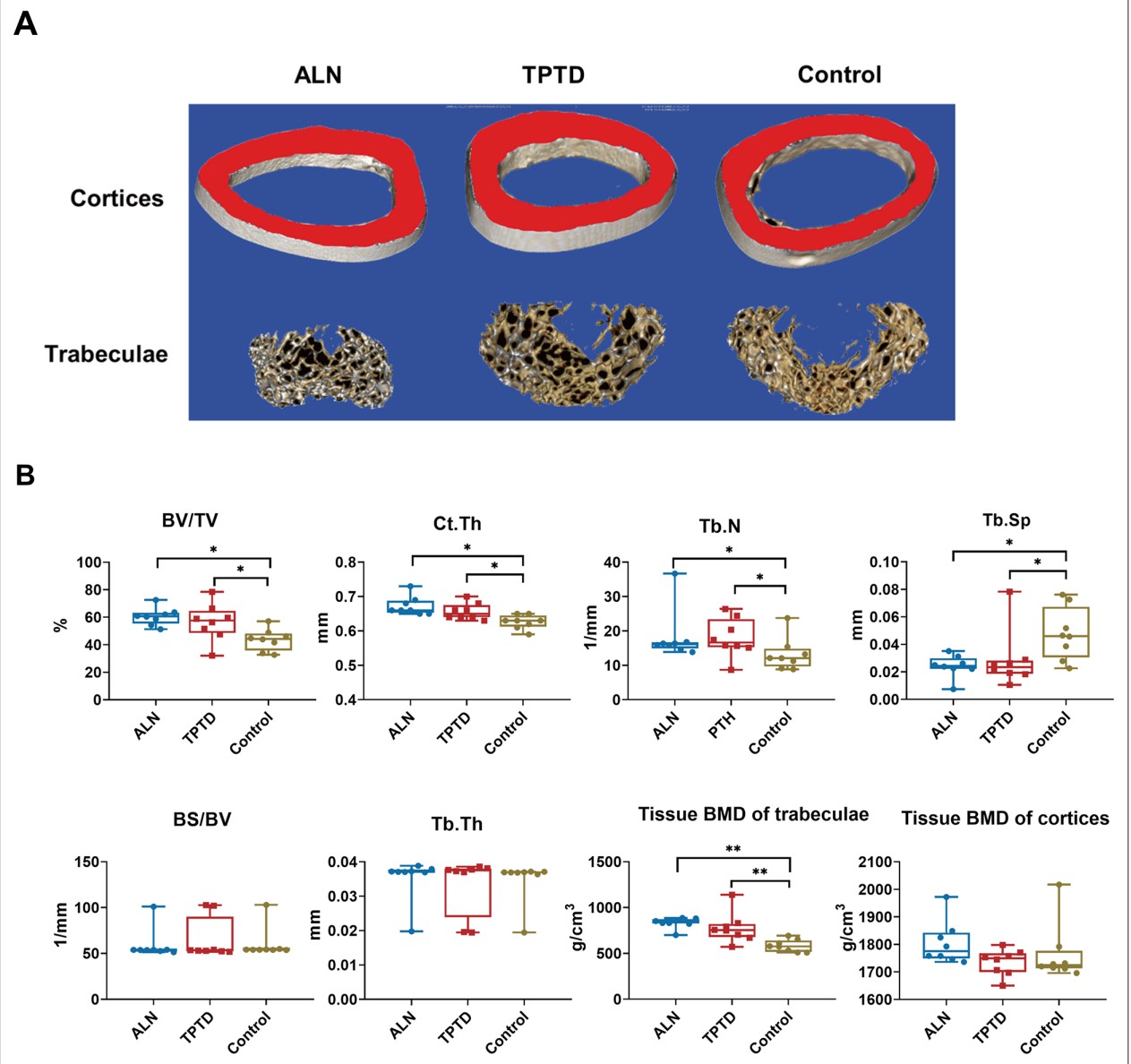

**Figure 3.** The efficacy of anti-osteoporotic treatment in *PLS3*[E10-16del/0] rats. (**A**) Three-dimensional reconstruction images of femurs after treatment. (**B**) Microstructural parameters of femurs by micro-CT after treatment (n=8 per group). ALN: alendronate, TPTD: teriparatide, BV/TV: bone volume/tissue volume, BS/BV: bone surface area/bone volume, Tb.Th: trabecular thickness, Tb.N: trabecular number, Tb.Sp: trabecular separation, Ct.Th: cortical thickness, BMD: tissue bone mineral density. Data were shown as the mean ± SD, evaluated by one-way ANOVA followed by Tukey's post hoc test. *p<0.05; **p<0.01; ***p<0.001.

The online version of this article includes the following source data for figure 3:

**Source data 1.** Microstructural parameters of femurs measured by micro-CT after treatment.

significant histomorphometric increase by 31.7% and 60.3% in %Tb.Ar and Tb.Th of $L_4$ was observed in TPTD group (all p<0.05 vs VEH group), but not in ALN group (*Figure 4A and B*). Tibial Ec.MAR of TPTD group was higher than VEH group (p<0.01), and the lowest Ec.MAR (2.75 μm/day) in the tibia was found in rats of ALN group (*Figure 4E*). Ps.MAR of the tibia was similar among ALN, TPTD, and VEH group (*Figure 4—figure supplement 1*).

Moreover, TPTD treatment significantly increased maximum load (182.2±8.7 N vs 154.2±15.9 N, p<0.001), yield load (117.5±22.8 N vs 87.8±16.6 N, p<0.05), and breaking load (165.3±9.5 N vs 142.2±20.8 N, p<0.01) of femur and maximum load of the fifth lumbar vertebrae (69.5±18.3 N vs

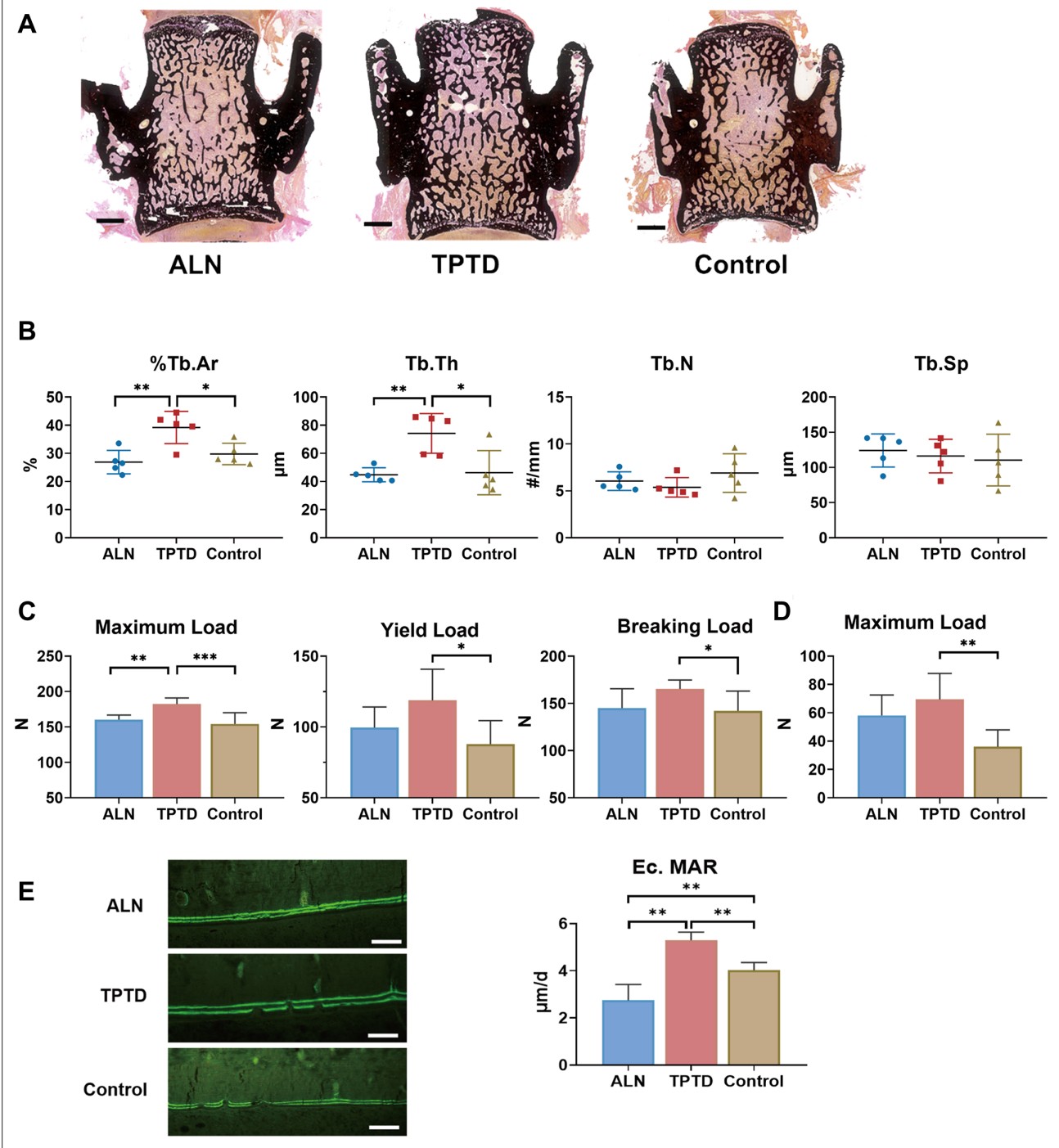

**Figure 4.** Changes in microarchitecture and strength after anti-osteoporotic treatment. (**A**) Typical images of unstained and uncalcified vertebra of *PLS3^{E10-16del/0}* rats after treatment. Scale bar = 1000 µm. (**B**) Histomorphometric analysis of L4 after treatment (n=5 per group). %Tb.Ar: trabecular area, Tb.Th: trabecular thickness, Tb.N: trabecular number, Tb.Sp: trabecular separation. (**C**) Effects of treatment on the mechanical strength of femoral diaphysis (n=8 per group). The diaphysis was subjected to three-point bending test to failure, which provided data on yield load, maximum load, breaking load. (**D**) Effects of treatment on the mechanical strength of L5 (n=7-8 per group). The vertebral body was subjected to indentation test to acquire maximum load. (**E**) Comparison of Ec.MAR in the tibial cortex among three treatment groups (n=5 per group). Scale bar = 100 µm. ALN: alendronate, TPTD: teriparatide, control: saline. Ec.MAR: mineral apposition rate of endocortical surface of tibia cortex. Data were shown as the mean ± SD, evaluated by one-way ANOVA followed by Tukey's post hoc test. *p<0.05; **p<0.01; ***p<0.001.

The online version of this article includes the following source data and figure supplement(s) for figure 4:

**Source data 1.** Changes in strength after anti-osteoporotic treatment.

**Figure supplement 1.** Characteristics of *PLS3^{E10-16del/0}* rats before and after treatment.

32.1±5.1 N, p＜0.01) than VEH group (*Figure 4C and D*). The bone strength of femur and lumbar vertebrae was not significantly improved in ALN group.

## Discussion

Recent advancements in genetic research have uncovered that loss-of-function variants in *PLS3* can cause a monogenic X-linked EOOP and osteoporotic fractures, but the exact mechanism is unknown (*Wolff et al., 2021*), and its optimal treatment regimen has not been established. In the present study, we have generated a novel rat model with large fragment deletion in *PLS3*, and we systematically assessed bone microarchitecture, bone biomechanical property, BMD, and bone remodeling for the first time in this novel rat model with patient-derived *PLS3* mutation. Interestingly, the newly generated *PLS3*$^{E10-16del/0}$ rat model displayed significantly impaired bone strength, decreased Ct.Th, and decreased MAR of trabecular bone. We demonstrated that ALN or TPTD could significantly improve bone microstructure and increase BMD, and TPTD could obviously improve the bone strength of *PLS3*$^{E10-16del/0}$ rat.

In this study, *PLS3*$^{E10-16del/0}$ rats displayed a bone-specific phenotype, including significantly impaired bone strength, decreased cortical bone thickness, and decreased MAR, despite the ubiquitous presence of the mutation. Similar results were found in patients with various *PLS3* mutations and the *PLS3*-deficient mice model (*Costantini et al., 2018*; *Fahiminiya et al., 2014*; *Hu et al., 2020*; *Kämpe et al., 2017*; *Kannu et al., 2017*; *Laine et al., 2015*; *Lv et al., 2017*; *Neugebauer et al., 2018*; *van Dijk et al., 2013*; *Wang et al., 2020*; *Yorgan et al., 2020*), which demonstrated that PLS3 had indispensable roles in bone metabolism. Interestingly, although we observed no change in bone mass in *PLS3*$^{E10-16del/0}$ rats, impaired bone strength was a prominent feature of this novel rat model, which could be attributed to the following factors. First, cortical wall thickness was significantly decreased in the *PLS3*$^{E10-16del/0}$ rats, which was a fundamental structural determinant of bone strength. Second, higher porosity in cortical bone and disorganized collagen fibers in bone played important roles in impairment of bone strength (*Gastaldi et al., 2020*).

PLS3 is expressed in all solid tissues except hematopoietic cells and is involved in all the processes dependent on filamentous actin (F-actin) dynamics. In bone, we also verified that PLS3 was widely expressed in osteoblasts, osteoclasts, and osteocytes (*Fahiminiya et al., 2014*; *Kamioka et al., 2004*; *Neugebauer et al., 2018*). Bone histomorphometric analysis indicated that the quantities of bone cells were normal in *PLS3*$^{E10-16del/0}$ rats, which was consistent with previous reports (*Neugebauer et al., 2018*; *Yorgan et al., 2020*). However, dysfunction of bone cells in *PLS3* mutant animal models and patients was found in the current and previous studies. Animal studies indicated PLS3 involvement in cytoskeletal actin bundling (*Oprea et al., 2008*), thus mediating mechanotransduction in osteocytes and further affecting cellular signal transduction between osteoblasts and osteoclasts, though the regulation was not confirmed (*Pathak et al., 2020*; *van Dijk et al., 2013*). Studies on patients' bone biopsies collectively insinuated a role for PLS3 in bone matrix mineralization (*Balasubramanian et al., 2018*; *Kämpe et al., 2017*; *Kannu et al., 2017*; *Laine et al., 2015*). Impaired bone mineralization induced by *PLS3* mutation was also observed in cell experiments and a murine model (*Fahiminiya et al., 2014*; *Yorgan et al., 2020*). F-actin-bundling ability or Ca$^{2+}$ sensitivity would be disturbed after *PLS3* mutations (*Schwebach et al., 2020*), which would affect intracellular calcium concentrations that was needed during osteoblasts differentiation and bone formation (*Wang et al., 2018*). In *PLS3*$^{E10-16del/0}$ rats, though immunohistochemical staining showed no significant changes in the expressions of Ocn, which was a late osteoblastic differentiation marker but did not represent a degree of bone mineralization (*Moriishi et al., 2020*), we found an obvious reduction in MAR of trabecular bone, indicating PLS3 was more likely involved in regulating bone mineralization. More recently, experimental findings suggested the regulatory function of PLS3 in osteoclastogenesis and osteoclast function through influencing podosome organization (*Neugebauer et al., 2018*). Taken together, these observations indicated that dysfunctional F-actin dynamics caused by *PLS3* mutation would lead to abnormal bone metabolism. However, the exact roles of *PLS3* regulating bone cells still needed to be further elucidated. Compared to the previously reported mice model with an entire deletion of *PLS3* (*Neugebauer et al., 2018*; *Yorgan et al., 2020*), *PLS3*$^{E10-16del/0}$ rats had a milder bone phenotype possibly due to the presence of truncated PLS3. PLS3 is comprised of an N-terminal Ca$^{2+}$-binding regulatory domain (RD) followed by a core consisting of two actin-binding domains. F-actin bundling by PLS3 is tightly regulated by Ca$^{2+}$ binding to RD (*Schwebach et al., 2020*). Deletion of *PLS3* E10-16

might result in a truncated protein with the RD retained as we detected the presence of PLS3 E1-9 cDNA. However, further studies are needed to verify the findings and explore the functional roles of different domains in bone regulation.

Recently, studies indicated that patients with *PLS3* mutations had significantly elevated serum DKK1 concentrations than patients with *WNT1* mutations, which indicated impaired WNT signaling may involve in the occurrence of *PLS3*-related osteoporosis (*Mäkitie et al., 2020a*). *PLS3*-deficient murine model exhibited decreased expression of *WNT16* (*Yorgan et al., 2020*). *WNT16*-deficient mice displayed cortical bone defects with normal trabecular bone, with unchanged cortical MAR (*Movérare-Skrtic et al., 2014*). Both *PLS3* and *WNT16* could inhibit osteoclasts formation by inhibiting NF-κB activation and *Nfatc1* expression (*Movérare-Skrtic et al., 2014*; *Neugebauer et al., 2018*). Therefore, *PLS3* mutation could lead to increased osteoclasts differentiation through decreased *PLS3* and *WNT16* expression, which could lead to thin cortical bone. However, serum bone turnover biomarker levels of ALP and β-CTX were normal in this *PLS3*$^{E10-16del/0}$ rats and patients with *PLS3* mutations (*Balasubramanian et al., 2018*; *Fahiminiya et al., 2014*; *Kämpe et al., 2017*; *Laine et al., 2015*), so more in-depth research on pathogenesis of *PLS3*-related osteoporosis was needed in the future.

Since patients with *PLS3* mutation presented with low BMD, multiple peripheral fractures, and vertebral compression fractures (*Hu et al., 2020*; *Lv et al., 2017*; *Mäkitie et al., 2020b*; *van Dijk et al., 2013*), it is of great clinical significance to establish an effective treatment regimen for *PLS3*-related osteoporosis. Bisphosphonates are still recommended as the first-line antiresorptive therapy for osteoporosis. TPTD, a recombinant fragment of human parathyroid hormone (1–34), is an extensively used bone anabolic drug for osteoporosis. It is noticed that the action mechanisms are completely different between antiresorptive and osteoanabolic agents. ALN inhibits osteoclast activity and bone resorption mainly via the mevalonate pathway (*Cremers et al., 2019*). Moreover, TPTD stimulates bone formation by affecting protein kinases, MAP-kinase, phospholipases, as well as the WNT signaling pathway (*Canalis et al., 2007*). In this study, ALN and TPTD were all effective in increasing BMD and improving bone microstructure of *PLS3*$^{E10-16del/0}$ rats, which were consistent with their efficacy in patients with PLS3-related EOOP (*Fratzl-Zelman et al., 2021*; *Hu et al., 2020*; *Lv et al., 2017*; *Välimäki et al., 2017*; *van Dijk et al., 2013*). Interestingly, TPTD treatment obviously improved the bone mechanical strength of femora and vertebrae of *PLS3*$^{E10-16del/0}$ rats, consistent with the improvement of bone microstructure of the above sites. However, we did not observe that ALN significantly improved bone biomechanical properties of *PLS3*$^{E10-16del/0}$ rats, which may be related to the small therapeutic dose and the short treatment time in this study. In another study, treatment with ALN at a dose of 30 μg/kg/day for 12 weeks significantly improved stiffness of a murine model of osteogenesis

**Table 1.** Small guide RNA (gRNA) sequence for *PLS3* gene knockout.

| No. | Targeting site | Name | Sequence (5'–3') |
|---|---|---|---|
| | | *PLS3*-gRNA-UP1 | TAGGatgtattctggcgccatct |
| 1 | ATGTATTCTGGCGCCATCT TGG | *PLS3*-gRNA-DOWN1 | aaacAGATGGCGCCAGAATACAT |
| | | *PLS3*-gRNA-UP2 | TAGGtacatacatacatagatga |
| 2 | TACATACATACATAGATGA TGG | *PLS3*-gRNA-DOWN2 | aaacTCATCTATGTATGTATGTA |
| | | *PLS3*-gRNA-UP3 | TAGGtctcaggtgaagtgcaca |
| | | *PLS3*-gRNA-down3 | aaacTGTGCACTTCACCTGAGA |
| | | *PLS3*-gRNA-UP4 | TAGGATCCAGGTCTACAGGAAAG |
| 3 | TCTCAGGTGAAGTGCACA TGG | *PLS3*-gRNA-down4 | aaacctttcctgtagacctggat |
| | | *PLS3*-gRNA-UP5 | TAGGATCCAGGTCTACAGGAAAG |
| 4 | CCT CTTTCCTGTAGACCTGGAT | *PLS3*-gRNA-down5 | AAACCTTTCCTGTAGACCTGGAT |

The gRNAs were designed based on *Rattus norvegicus* (Norway rat) genome assembly Rnor_6.0 (rn6), using CRISPR Design Tool (http://tools.genome-engineering.org).

Gene ID: 81748; Location: Chromosome X - NC_005120.4.

The words in red represented the protospacer adjacent motif (PAM) sequences (5'–3').

imperfecta (*McCarthy et al., 2002*). It is necessary to carry out studies with a larger dose and longer time treatment of ALN to clarify its effects on bone strength of *PLS3*$^{E10\text{-}16del/0}$ rats.

Although the precise molecular mechanisms of *PLS3* mutation inducing EOOP needed to be clarified, we confirmed that *PLS3* played critical roles in regulating bone metabolism and maintaining the integrity of bone structure and mechanical properties in this novel *PLS3*$^{E10\text{-}16del/0}$ rat model. Our results demonstrated for the first time that TPTD and ALN were effective to *PLS3*$^{E10\text{-}16del/0}$ rats, which provided valuable experimental evidence for the treatment of *PLS3*-related EOOP. However, there were a few limitations. Since *PLS3* was widely expressed, we did not conduct enough morphometrical and functional analyses of other tissues and organs of *PLS3*$^{E10\text{-}16del/0}$ rats. We did not conduct in-depth research on the signal pathways regulating bone metabolism, such as WNT/β-catenin, OPG-RANK-RANKL pathway, and so on in *PLS3*$^{E10\text{-}16del/0}$ rats, which were helpful to clarify the pathogenesis of this EOOP.

In conclusion, *PLS3* plays major roles in bone metabolism and bone integrity. Impaired bone microstructure and bone strength were prominent characteristics of rats with hemizygous E10-16del mutation of *PLS3*, though the exact pathogenesis still needs further study. ALN and TPTD treatment can increase BMD and improve the bone microstructure of rats with *PLS3*-related EOOP.

## Materials and methods
### Generation of *PLS3* KO rat model and genetic identification

General *PLS3* KO rat model was generated using the CRISPR/Cas9 system at the Institute of Laboratory Animal Sciences, Chinese Academy of Medical Sciences & Peking Union Medical College. Five specific guide RNAs were used to target the exons 10–16 of *PLS3* (*Table 1*), which were annealed and cloned into the PUC57-gRNA expression vector (Addgene 51132, Cambridge, MA, USA) with a T7 promoter. In vitro, transcription of gRNA template was accomplished using the MEGAshortscript Kit (AM1354, Ambion). The pST1374-NLS-flag-linker-Cas9 vector (Addgene 44758) was linearized using the Age I enzyme and transcribed with a T7 Ultra Kit (Ambion, AM1345). After purified with the MEGAclear Kit (AM1908, Ambion), a mixture of transcribed Cas9 mRNA and gRNA was microinjected into the cytoplasm of zygotes of Sprague Dawley (SD) rats which were obtained commercially from

**Table 2.** Primers used for Sanger sequencing, genotyping, and qPCR.

| Primer name | Sequence (5'–3') | GenBank accession number | Nucleotide position |
|---|---|---|---|
| P-WT-F | CCCATAAGTTGTTCCTTGATTTCC | NC_051356.1 | 88350–88373 |
| P-WT-R | CACTGCCTGAATAAGACCCACTC | NC_051356.1 | 88777–88799 |
| P-WT-F' | CCTTGTGGAAGTAAAACCGAA | NC_051356.1 | 93529–93549 |
| P-WT-R' | TACAAAGGCCAAGTTCAG | NC_051356.1 | 84492–84510 |
| P-KO-F | CTTCATTCCCTTTGCACGTT | NC_051356.1 | 84092–84111 |
| P-KO-R | TACAAAGGCCAAGTTCAG | NC_051356.1 | 94242–94262 |
| *PLS3*-KO-F | AAATTCTCCTTGGTTGGCATT | NM_031084.1 | 1507–1527 |
| *PLS3*-KO-R | TCCAGCTTCACTCAATGTTCC | NM_031084.1 | 1672–1692 |
| *PLS3*-F | GAAAATGATCCCGATTGCAG | NM_031084.1 | 496–515 |
| *PLS3*-R | CTCTCATCAATTGTATCGGGAA | NM_031084.1 | 605–626 |
| *COL1A1*-F | TCCTGACGCATGGCCAAGAA | NM_053304.1 | 164–183 |
| *COL1A1*-R | CATAGCACGCCATCGCACAC | NM_053304.1 | 289–308 |
| *GAPDH*-F | TTCAACGGCACAGTCAAGG | NM_017008.4 | 235–253 |
| *GAPDH*-R | CTCAGCACCAGCATCACC | NM_017008.4 | 331–348 |

Primers *PLS3*-KO-F/*PLS3*-KO-R were selected to amplify a fragment spanning from exon 13 to exon 14 of the rat *PLS3* cDNA. The qPCR results were used to indicate the expression level of *PLS3* E10-16 (knockout region). Primers *PLS3*-F/*PLS3*-R were selected to amplify a fragment spanning from exon 5 to exon 6 of the rat *PLS3* cDNA. The qPCR results were used to indicate the expression level of *PLS3* E1-9 (uneditable region).

F: forward, R: reverse, WT: wild-type, KO: knockout.

Beijing Vital River Laboratory Animal Technology Co., Ltd. A male founder was backcrossed with WT rats to generate female heterozygous KO rats (*PLS3*$^{E10-16del/+}$), which were subsequently crossed with WT male rats. As being located on the X chromosome, *PLS3*-induced osteoporosis was more severe in males than females, male hemizygous KO rats (*PLS3*$^{E10-16del/0}$) of the offspring were selected to be further studied. Since *PLS3*-related osteoporosis had its disease onset early in childhood and adolescence, and rats reached the peak bone mass at 6–9 months of age, we chose rats at age of 3, 6, and 9 months for further investigation. All rats were registered and preserved on an SD genetic background in a specific pathogen-free environment without surpassing six animals in ventilated cages. Rats were provided with standard chow and water ad libitum.

Genomic DNA was isolated from tail snips using E.Z.N.A. Tissue DNA Kit (Omega Bio-tek, Norcross, GA, USA). Genotyping was performed using PCR amplification. The allele-specific primers were listed in *Table 2*. Primers P-KO-F/P-KO-R were used to amplify *PLS3* KO allele, and a 554 bp PCR production would be generated. Primers P-WT-F/P-WT-R were used to amplify WT allele, and a 450 bp PCR production would be generated. Thermal cycling conditions consisted of an initial denaturation at 95°C for 3 min, followed by 38 cycles at 95°C for 30 s, 52–58°C for 30 s, and 72°C for 1 min. Sanger sequencing of PCR products was further completed.

## Western blot analysis

In order to confirm the general deletion of PLS3 protein, proteins from bone, muscle, heart, lung, renal, and liver of *PLS3*$^{E10-16del/0}$ and WT rats were extracted for western blot. Briefly, the tissues were separately homogenized in RIPA buffer containing 1 mM PMSF and Halt Protease and Phosphatase Inhibitor cocktail (Thermo Fisher Scientific, Waltham, MA, USA). The protein concentration was determined with BCA Protein Assay Kit (Thermo Fisher Scientific). Protein samples (30 µg each) were loaded onto a 4–12% gradient SDS-PAGE gel and then transferred to PVDF membranes. After being blocked with 5% BSA, the membranes were incubated with primary antibody against PLS3 (Cat# ab233104, Abcam, 1:1000 dilution) or GAPDH (Cat# ab8245, Abcam, 1:1000 dilution) and washed with TBST three times, which were then incubated with a horseradish peroxidase-conjugated anti-rabbit or anti-mouse secondary antibody. Finally, the immunoblots were visualized by enhanced chemiluminescence.

## Real-time qPCR

Total RNA was extracted from the tibia of WT and *PLS3*$^{E10-16del/0}$ rats with TRIzol reagent and was reversely transcribed to cDNA using the PrimeScript RT Reagent Kit (Takara, Kusatsu, Japan). The gene expression levels were quantified by qPCR using TB Green Premix Ex Taq II (Tli RNase H Plus, Takara) on a Viia 7 Real-Time PCR System (Life Technologies, USA). Primer sequences for RT-qPCR were listed in *Table 2*. Relative mRNA expression levels were calculated using the $2^{-\Delta\Delta CT}$ method and normalized to the internal control *GAPDH*.

## Treatment

A total of 24 *PLS3*$^{E10-16del/0}$ male rats at 3 months of age were randomized either to VEH, ALN, or TPTD therapy (n=8 in each group) for 8 weeks. ALN (Merck and Co., Inc, Rahway, NJ, USA) 1.0 µg/kg body weight was subcutaneously injected daily into rats in ALN group (*Diab et al., 2011*; *Iwata et al., 2006*). In the TPTD group, the agent (recombinant human parathyroid hormone 1–34, provided by Salubris Biotherapeutics, Inc, Shenzhen, China) was subcutaneously injected at the dose of 40 µg/kg body weight, five times weekly (*Komrakova et al., 2010*; *Lane et al., 1996*). Rats in the VEH group received 0.9% saline and acted as control groups.

All animal experiments were approved by the Institutional Animal Care and Use Committee of the Peking Union Medical College Hospital (XHDW-2021-027).

## Bone microstructure assessment

The left femur was fixed in 4% paraformaldehyde, of which microarchitecture was assessed by micro-computed tomography (µCT) (Inveon MM CT, Siemens, Erlangen, Germany) according to the recommended protocol (*Bouxsein et al., 2010*). In vitro scans were operated with an X-ray tube voltage of 60 kV, a current of 400 µA, an exposure time of 800 ms, and a voxel size of 20 µm. The region of interest for trabecular bone was drawn in the distal epiphysis, starting 1.5 mm below the growth plate and extending 100 slices to proximal end. Cortical bone was analyzed in a 1000-µm-long volume

situated in the middle of the diaphysis. BMD, BV/TV, BS/BV, Ct.Th, Tb.Th, Tb.N, and Tb.Sp were measured. The Inveon Research Workplace software (Siemens) was used for reconstruction and analysis of two-dimensional (2D) and 3D image.

## Assessment of biomechanical properties of bone

The right femur and the fifth lumbar vertebrae ($L_5$) were wrapped in saline-soaked gauze and stored at –20°C, which were thawed at room temperature 2 hr before the mechanical test. Three-point binding tests and indentation testing were implemented on a fatigue-testing machine (BOSE ElectroForce 3200, TA Instruments, New Castle, DE, USA). From the load-displacement curves of femurs, stiffness, yield load, maximum load, breaking load, post-yield displacement, and work-to-fracture were generated. Force-displacement measurement was also performed on the fifth lumbar vertebral bodies, and maximum load was measured.

## Measurement of bone metabolic markers and metabolic parameters

Under abdominal anesthesia, blood samples were collected via cardiac puncture. Concentrations of serum calcium (Ca), phosphorus (P), and ALP (a bone formation marker) were measured by an automated chemistry analyzer (AU5800, Beckman Coulter Inc, Brea, CA, USA). Serum level of C-telopeptide of type Ⅰ collagen (β-CTX, bone resorption marker) was measured by ELISA (Cat# CSB-E12776r, Cusabio Biotech Co., Wuhan, China). To determine whether *PLS3* KO affected other metabolism, we measured the levels of serum Glu, TC, LDL, and TG using the automated chemistry analyzer.

## Analysis of histology and histomorphometry

All rats received intraperitoneal injection of calcein (10 mg/kg body mass, Sigma-Aldrich, Co., St. Louis, MO, USA) for histomorphometric analysis on the second and the sixth day before euthanization. The left femur was decalcified after μCT scanning and were embedded in paraffin and cut into 4 μm sections using a microtome (Leica RM2016, Leica Microsystems). H&E staining and tartrate-resistant acid phosphatase staining (Servicebio, Cat# G1050) were performed. The osteoclast number per bone perimeter (N.Oc/B.Pm), osteocyte number per bone area (N.Ot/B.Ar), and osteoblast number/bone perimeter (N.Ob/B.Pm), number and area of adipocytes in the distal marrow per tissue area were calculated. Bone slices were also subjected to the picrosirius red staining for the evaluation of the organization of collagen fibers through polarized light microscopy (Axio Imager D2, Zeiss, Germany).

The right tibia and the fourth vertebrae ($L_4$) were fixed in 70% alcohol and embedded in modified methyl methacrylate without decalcification. The embedded samples were cut into 10 μm thick sections, which were de-plasticized and stained with von Kossa stain kit (Servicebio, Cat# G1043) to calculate trabecular area (%Tb.Ar), Tb.Th, Tb.N, and Tb.Sp. For unstained slices, the MAR was calculated by dividing the distance between the two calcein labels by the inter-labeling period. Analysis was performed with ImageJ software according to the recommendation of ASBMR (*Dempster et al., 2013*).

## Immunohistochemical staining

Paraffin-embedded femur samples were subjected to immunostaining for PLS3 and osteocalcin (Ocn). Slides were subjected to 0.05% trypsin at 37°C for 30 min for antigen retrieval and blocked in 3% hydrogen peroxide and 3% bovine serum albumin. Rabbit anti-PLS3 antibody (Cat# ab23310, Abcam, 1:200 dilution) or anti-Ocn antibody (Cat# ab93876, Abcam, 1:200 dilution) were applied, followed by incubation with HRP-labeled secondary antibody (ZSGB-BIO). PLS3 and Ocn expression were visualized using 3,3'-diaminobenzidine staining. Counterstaining was performed using hematoxylin. Ocn-positive area was quantified using ImageJ software.

## Statistical analysis

All experiments were repeated at least thrice independently. Results were reported as means ± SD (standard difference). Comparisons of parameters between transgenic and WT rats were completed using Student's t test. Parameters of rats at different ages or in different treatment groups were compared with one-way ANOVA followed by Tukey's post hoc test. Statistical analysis was performed using SPSS Statistics 26.0 (IBM, Armonk, NY, USA), GraphPad Prism 8 (Statcon). Statistical significance was determined when p values equal to or less than 0.05.

## Acknowledgements

This work is supported by National Key R&D Program of China (2018YFA0800801, 2021YFC2501704), CAMS Innovation Fund for Medical Sciences (CIFMS) (2021-I2M-C&T-B-007, 2021-I2M-1-051), National Natural Science Foundation of China (No. 81873668, 82070908), Beijing Natural Science Foundation (7202153), and the Fundamental Research Funds for the Central Universities (3332022102).

## Additional information

### Funding

| Funder | Grant reference number | Author |
| --- | --- | --- |
| National Key Research and Development Program of China | 2018YFA0800801 | Mei Li |
| Chinese Academy of Medical Sciences Initiative for Innovative Medicine | 2021-I2M-C&T-B-007 | Mei Li |
| National Natural Science Foundation of China | No.81873668 | Mei Li |
| Beijing Natural Science Foundation | 7202153 | Mei Li |
| Fundamental Research Funds for the Central Universities | 3332022102 | Jing Hu |
| National Key Research and Development Program of China | 2021YFC2501704 | Mei Li |
| Chinese Academy of Medical Sciences Initiative for Innovative Medicine | 2021-I2M-1-051 | Mei Li |
| National Natural Science Foundation of China | 82070908 | Mei Li |

The funders had no role in study design, data collection and interpretation, or the decision to submit the work for publication.

### Author contributions

Jing Hu, Conceptualization, Investigation, Visualization, Writing – original draft; Bingna Zhou, Conceptualization, Investigation, Visualization; Xiaoyun Lin, Lei Sun, Jiayi Liu, Conceptualization, Investigation; Qian Zhang, Investigation, Methodology; Feifei Guan, Resources, Investigation, Methodology; Ou Wang, Yan Jiang, Resources, Supervision; Wei-bo Xia, Resources, Writing - review and editing; Xiaoping Xing, Writing - review and editing; Mei Li, Conceptualization, Supervision, Funding acquisition, Writing – original draft

### Author ORCIDs

Mei Li http://orcid.org/0000-0002-4380-3511

### Ethics

This study was performed in strict accordance with the recommendations in the Guide for the Care and Use of Laboratory Animals of the National Institutes of Health. All animal experiments were approved by the Institutional Animal Care and Use Committee of the Peking Union Medical College Hospital (XHDW-2021-027). Every effort was made to minimize pain and suffering by providing support when necessary and choosing ethical endpoints.

### Decision letter and Author response

Decision letter https://doi.org/10.7554/eLife.80365.sa1
Author response https://doi.org/10.7554/eLife.80365.sa2

## Additional files

### Supplementary files
• MDAR checklist

### Data availability
All data analyzed during this study are included in the manuscript and supporting file. Source Data files have been provided for Figures 1-4.

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
