## [Editor Report]

The findings are significant with regard to the clinical treatment of early onset osteoporosis due to PLS3 mutations. The evidence which supports the conclusions of the manuscript is strong. This manuscript will be of notable relevance to the field of metabolic bone diseases.

---

## [Decision Letter]

**Decision letter after peer review:**

Thank you for submitting your article "Impaired bone strength and bone microstructure in a novel early-onset osteoporotic rat model with a clinically relevant *PLS3* mutation" for consideration by *eLife*. Your article has been reviewed by 2 peer reviewers, and the evaluation has been overseen by a Reviewing Editor and Mone Zaidi as the Senior Editor. The reviewers have opted to remain anonymous.

Essential revisions:

1) Please provide additional information about the detection of the knockdown efficiency with PLS3 protein expression in bone samples and in other tissues.

2) In the H&E stains of the distal femur of PLS3E10-16del/0 rats please quantify adipocytes and either OCN or Runx2 with immunohistochemical staining.

3) Please explain how the bone phenotype for this knockout differs from Neuberger et al. 2018 and Yorgan et al. 2020.

4) Please explain if there were differences between WT and KO in organs or metabolism including life expectancy.

5) Please explain why bone strength is lower in KO vs WT when bone mass does not differ and whether this could be related to bone quality.

*Reviewer #1 (Recommendations for the authors):*

In their paper, Hu et al. constructed a novel rat model with a clinically relevant PLS3 hemizygous E10-16del mutation (PLS3E10-16del/0), which presents a classic form of early-onset osteoporosis. Concomitant treatment with the clinical drugs alendronate and teriparatide significantly improved bone mass and bone microarchitecture. The research on this topic has certain important significance for the clinical treatment of early-onset osteoporosis caused by PLS3 mutation.

This is paper is of particular interest as it, in an elegant way, is based on a series of relevant in vivo experiments. Results are based on straightforward experiments which allow the authors to draw solid conclusions. The paper is well-organized and comprehensibly written. Some linguistic revision is recommended.

1. Insufficient experimental data for establishment and evaluation of PLS3E10-16del/0 rats.

Please supplement PLS3E10-16del/0 rats and their wild-type littermates were genotyped by DNA sequencing results and related experiments to supplement the detection of knockdown efficiency: PCR and WB or immunofluorescence detection of PLS3 gene or protein in bone samples. What's more, if possible, collect other tissues and organs of rats for analysis of PLS3 protein expression.

2. It is recommended to supplement the H&E staining of the distal femur of PLS3E10-16del/0 rats to quantify the number and area of adipocytes and to analyze the expression of OCN or Runx2 osteogenesis-related proteins by immunohistochemical staining.

*Reviewer #2 (Recommendations for the authors):*

The authors generated the PLS3 rat model and performed a nice set of analyses to confirm its relevance to early-onset osteoporosis (EOOP). In addition, the authors also tested some anti-osteoporosis drugs in this PLS3 model and show their efficacy in increasing bone mass and bone formation. A few points for the author to consider for improving the manuscript:

(1) Page 4, 2nd paragraph: It will be helpful for readers if a brief description can be added to describe the differences between the current knockout in this paper vs. Neugebauer et 8 al., 2018; Yorgan et al., 2020 for the bone phenotype.

(2) Body weight did not differ between WT and KO. Is there any difference in any other organs or metabolism including life expectance between KO vs. WT?

(3) Fig 2 and suppl Fig 2: Overall, the difference in bone mass (BV/TV) is not significant between the KO vs. WT in different age groups. However, parameters related to bone strength are significantly lower in KO vs. WT. Can the authors please provide some possible explanations? Maybe the poor bone quality contributes to the lower bone strength in the KOs? Is there any way the authors can check the bone quality-related parameters?

---

## [Author Response]

Reviewer #1 (Recommendations for the authors):In their paper, Hu et al. constructed a novel rat model with a clinically relevant PLS3 hemizygous E10-16del mutation (PLS3E10-16del/0), which presents a classic form of early-onset osteoporosis. Concomitant treatment with the clinical drugs alendronate and teriparatide significantly improved bone mass and bone microarchitecture. The research on this topic has certain important significance for the clinical treatment of early-onset osteoporosis caused by PLS3 mutation.This is paper is of particular interest as it, in an elegant way, is based on a series of relevant in vivo experiments. Results are based on straightforward experiments which allow the authors to draw solid conclusions. The paper is well-organized and comprehensibly written. Some linguistic revision is recommended.1. Insufficient experimental data for establishment and evaluation of PLS3E10-16del/0 rats.Please supplement PLS3E10-16del/0 rats and their wild-type littermates were genotyped by DNA sequencing results and related experiments to supplement the detection of knockdown efficiency: PCR and WB or immunofluorescence detection of PLS3 gene or protein in bone samples. What's more, if possible, collect other tissues and organs of rats for analysis of PLS3 protein expression.

Thank you for the valuable comments. Based on your suggestions, some further experiments were completed. We use DNA sequencing, western blotting, real-time quantitative PCR, and immunohistochemical staining to verify that a novel rat model with deletion of exons 10-16 in the rat *PLS3* gene was successfully built. The methods were described as follows:

“Genomic DNA was isolated from tail snips using E.Z.N.A. Tissue DNA Kit (Omega Bio-tek, Norcross, GA, US). Genotyping was performed using polymerase chain reaction (PCR) amplification. The allele-specific primers were listed in Table 2. Primers P-KO-F/P-KO-R were used to amplify *PLS3* knock-out (KO) allele, and a 554bp PCR production would be generated. Primers P-WT-F/P-WT-R were used to amplify wild-type allele, and a 450bp PCR production would be generated. Thermal cycling conditions consisted of an initial denaturation at 95 ℃ for 3 min, followed by 38 cycles at 95℃ for 30s, 52-58℃ for 30s, and 72℃ for 1 min. Sanger sequencing of PCR products was further completed.” (Page 14, Line 8-15)

“Western blot analysis

In order to confirm the general deletion of PLS3 protein, proteins from bone, muscle, heart, lung, renal, and liver of *PLS3^E10-16del/0^* and WT rats were extracted for western blot. Briefly, the tissues were separately homogenized in RIPA buffer containing 1 mM PMSF and Halt Protease and Phosphatase Inhibitor cocktail (Thermo Fisher Scientific, Waltham, USA). The protein concentration was determined with BCA Protein Assay Kit (Thermo Fisher Scientific). Protein samples (30 ug each) were loaded onto a 4-12% gradient SDS-PAGE gel and then transferred to PVDF membranes. After being blocked with 5% BSA, the membranes were incubated with primary antibody against PLS3 (Cat# ab233104, Abcam, 1:1000) or GAPDH (Cat#ab8245, Abcam, 1:1000 dilution) and washed with TBST three times, which were then incubated with a horseradish peroxidase-conjugated anti-rabbit or anti-mouse secondary antibody. Finally, the immunoblots were visualized by enhanced chemiluminescence.” (Page 14, Line 17-22, Page 15, Line 1-5)

“Real-time quantitative PCR

Total RNA was extracted from the tibia of WT and *PLS3^E10-16del/0^* rats with TRIzol reagent and was reversely transcribed to cDNA using the PrimeScript RT Reagent Kit (Takara, Kusatsu, Japan). The gene expression levels were quantified by qPCR using TB Green Premix Ex Taq II (Tli RNase H Plus, Takara) on a Viia 7 Real-Time PCR System (Life Technologies, USA). Primer sequences for RT-qPCR were listed in Table 2. Relative mRNA expression levels were calculated using the 2 −ΔΔCT method and normalized to the internal control *GAPDH*.” (Page 15, Line 7-12)

“Immunohistochemical staining

Paraffin-embedded femur samples were subjected to immunostaining for PLS3 and osteocalcin (Ocn). Slides were subjected to 0.05% trypsin at 37°C for 30 minutes for antigen retrieval and blocked in 3% hydrogen peroxide and 3% bovine serum albumin. Rabbit anti-PLS3 antibody (Cat#ab23310, Abcam, 1:200 dilution) or anti-Ocn antibody (Cat#ab93876, Abcam, 1:200 dilution) were applied, followed by incubation with HRP-labeled secondary antibody (ZSGB-BIO). PLS3 and Ocn expression were visualized using 3, 3'-diaminobenzidine (DAB) staining. Counterstaining was performed using hematoxylin. Ocn-positive area were quantified using Image-J software.” (Page 18, Line 4-12)

We summarized our findings in the result part as follows:

“A large fragment deletion of exon 10-16 in *PLS3* was introduced into the genome of rats, and a 9626-bp deletion from 84172 bp to 93797 bp (NC_051356.1) was confirmed by genotyping and Sanger sequencing (Figure 1A, 1B). PLS3 antibody used in this study targeted the knockout region of PLS3 protein (deleted amino acid region: 331-630). Therefore, using western blotting of equivalent amounts of total proteins from wild type (WT) and *PLS3^E10-16del/0^* rats, we observed a band of appropriate size for PLS3 (~70kDa) in membranes of WT rats, which was lacking in membranes of *PLS3^E10-16del/0^* rats (Figure 1C). Immunohistochemical staining of femoral sections also unraveled that PLS3 was present in osteocytes, osteoblasts, and osteoclasts in cortical and trabecular bone of WT rats, while PLS3 was not seen in bone cells of the *PLS3^E10-16del/0^* rats (Figure 1E). Moreover, qPCR results also indicated that the expression level of *PLS3 E10-16* was extremely low in *PLS3^E10-16del/0^* rats. However, we found a similar expression level of *PLS3* E1-9 between *PLS3^E10-16del/0^* and WT rats, which indicated a possible presence of truncated PLS3 variants (Figure 1D). Together, these results indicated that rats with hemizygous E10-16 deletion in *PLS3* were successfully built.” (Page 5, Line 1-14)

2. It is recommended to supplement the H&E staining of the distal femur of PLS3E10-16del/0 rats to quantify the number and area of adipocytes and to analyze the expression of OCN or Runx2 osteogenesis-related proteins by immunohistochemical staining.

Thank you for your valuable suggestions. We have quantified the number and area of adipocytes and analyzed the expression of Ocn by immunohistochemical staining.

The methods are described as follows:

“Hematoxylin & eosin (H&E) staining and tartrate-resistant acid phosphatase (TRAP) staining (Servicebio, Cat# G1050) were performed. The osteoclast number per bone perimeter (N.Oc/B.Pm), osteocyte number per bone area (N.Ot/B.Ar), and osteoblast number/bone perimeter (N.Ob/B.Pm), number and area of adipocytes in the distal marrow per tissue area were calculated. Bone slices were also subjected to the picrosirius red (PS) staining for the evaluation of the organization of collagen fibers through polarized light microscopy (Axio Imager D2, Zeiss, Germany).” (Page 17, Line 13-19)

“The method of immunohistochemical staining was described in our response to comment 1.” (Page 18, Line 4-12).

The corresponding results were described as follows:

“Also, *PLS3^E10-16del^*^/0^ and WT rats had similar Ocn-positive areas at the trabecular and cortical bone of femur (Figure 2—figure supplement 3A). Serum levels of total ALP, β-CTX and calcium were similar in *PLS3^E10-16del^*^/0^ and WT rats. (Figure 2—figure supplement 3C).” (Page 7, Line 1-4)

“No difference was found in the area of adipocytes in the distal marrow per tissue area. However, 3-month-old *PLS3^E10-16del^*^/0^ rats had more adipocytes than WT rats of the same age (Figure 2—figure supplement 3B).” (Page 7, Line 10-12)

Reviewer #2 (Recommendations for the authors):The authors generated the PLS3 rat model and performed a nice set of analyses to confirm its relevance to early-onset osteoporosis (EOOP). In addition, the authors also tested some anti-osteoporosis drugs in this PLS3 model and show their efficacy in increasing bone mass and bone formation. A few points for the author to consider for improving the manuscript:(1) Page 4, 2nd paragraph: It will be helpful for readers if a brief description can be added to describe the differences between the current knockout in this paper vs. Neugebauer et 8 al., 2018; Yorgan et al., 2020 for the bone phenotype.

Thank you for the valuable suggestions. We have added information about the differences in the bone phenotype between this novel rat model and the previously reported mice model with PLS3 knockout as follows:

So far, the animal models carrying patient-derived PLS3 mutations have not been generated, which are valuable to unveil the pathogenesis of this ultra-rare X-linked osteoporosis induced by PLS3 mutations. A previously reported PLS3 knock-out (KO) murine model displayed a significant decrease in cortical thickness with normal or decreased trabecular number. This model harbored a complete deletion of PLS3 gene that was different from mutations identified in patients (Neugebauer et al., 2018; Yorgan et al., 2020). Moreover, effective treatment strategies have not been established in EOOP related to PLS3 mutations. Few patients with PLS3-related EOOP received bisphosphonates or teriparatide treatment, while the efficacy was variable (Fratzl-Zelman et al., 2021; Hu et al., 2020; Lv et al., 2017; Valimaki et al., 2017; van Dijk et al., 2013). To explore the potential pathogenesis and treatment strategies of PLS3-related EOOP, we independently constructed a novel rat model with ubiquitous deletion of the exon 10-16 of PLS3 (PLS3E10-16del/0), which recapitulated a patient-specific mutation of exon 10-16 deletion in PLS3 (Lv et al., 2017). We observed that PLS3E10-16del/0 rats, similar to the respective patient, displayed impaired bone strength due to thinner cortical bone, which could be improved by treatment with alendronate and teriparatide. Compared to the previously reported mice model, the novel rat model had a milder bone phenotype possibly due to the presence of truncated PLS3 variants. (Page 4, Line 5-21)

Also, we discussed the differences in the discussion part as follows:

Compared to the previously reported mice model with an entire deletion of PLS3 (Neugebauer et al., 2018; Yorgan et al., 2020), PLS3E10-16del/0 rats had a milder bone phenotype possibly due to the presence of truncated PLS3. PLS3 is comprised of an N-terminal Ca2+-binding regulatory domain (RD) followed by a core consisting of two actin-binding domains. F-actin bundling by PLS3 is tightly regulated by Ca2+ binding to RD (Schwebach et al., 2020). Deletion of PLS3 E10-16 might result in a truncated protein with the regulatory domain retained as we detected the presence of PLS3 E1-9 cDNA. However, further studies are needed to verify the findings and explore the functional roles of different domains in bone regulation. (Page 10, Line 19-23. Page 11, Line 1-4)

(2) Body weight did not differ between WT and KO. Is there any difference in any other organs or metabolism including life expectance between KO vs. WT?

Thank you very much. We have added more information about the extra-skeletal phenotypes of PLS3E10-16del/0 rats and the life expectance were similar between the two groups. The additional experiments and related description were shown as follows:

Measurement of bone metabolic markers and metabolic parameters

Under abdominal anesthesia, blood samples were collected via cardiac puncture. Concentrations of serum calcium (Ca), phosphorus (P), and alkaline phosphatase (ALP, a bone formation marker) were measured by an automated chemistry analyzer (AU5800, Beckman Coulter Inc, Brea CA, USA). Serum level of C-telopeptide of type Ⅰ collagen (β-CTX, bone resorption marker) was measured by ELISA (Cat# CSB-E12776r, Cusabio Biotech Co., Wuhan, China). To determine whether PLS3 knock-out affected other metabolism, we measured the levels of serum Glu, TC, LDL, and TG using the automated chemistry analyzer. (Page 17, Line 4-5)

We described our findings as follows:

Moreover, the life span of WT and PLS3E10-16del/0 rats were similar and no rats died during the experimental period. There were also no statistical differences in body weight change between WT and PLS3E10-16del/0 rats during the whole experimental period (Figure 1- figure supplement 1). (Page 5, Line 18-21)

We also investigated the parameters of glucose and lipid metabolism and found similar levels of serum glucose (Glu), triglycerides (TG), cholesterol (TC), low-density lipoprotein (LDL) between PLS3E10-16del/0 rats and WT at all ages (Figure 2-figure supplement 3D). Besides, PLS3 mutations were reported to have patent ductus arteriosus (Qiu, Li, Zhang, & Liu, 2022). However, no obvious abnormalities were found in ultrasonic cardiogram and hematein eosin (HE)-staining about the cardiac structure and function of PLS3E10-16del/0 rats. (Page 7, Line 12-18)

(3) Fig 2 and suppl Fig 2: Overall, the difference in bone mass (BV/TV) is not significant between the KO vs. WT in different age groups. However, parameters related to bone strength are significantly lower in KO vs. WT. Can the authors please provide some possible explanations? Maybe the poor bone quality contributes to the lower bone strength in the KOs? Is there any way the authors can check the bone quality-related parameters?

Thank you for the valuable advice. PLS3E10-16del/0 rats displayed thinner cortical bone, which was a fundamental structural determinant of bone strength. In order to figure out more reason of impaired strength in femur and lumbar spine of PLS3E10-16del/0 rats, we also used qPCR to detect the expression of COL1A1, use picrosirius red staining to evaluate the content of collagen of bone tissue under polarized microscopy.

The methods of additional experiments are described as follows:

Bone slices were also subjected to the picrosirius red (PS) staining for the evaluation of the organization of collagen fibers through polarized light microscopy (Axio Imager D2, Zeiss, Germany). (Page 17, Line 15-17)

The results were described as follows:

Since no difference were found in bone mass between WT and PLS3E10-16del/0 rats, we further investigated the quantity and structure of bone collagen. The expression of COL1A1 in tibia were also similar between two groups. However, compared to WT rats, the cortical bone was more porous and the collagen fibers of PLS3E10-16del/0 rats were relatively disorganized (Figure 2H). (Page 7, Line 5-9).

We discussed the findings in our discussion part as follows:

Interestingly, although we observed no change in bone mass in PLS3E10-16del/0 rats, impaired bone strength was a prominent feature of this novel rat model, which could be attributed to the following factors. First, cortical wall thickness was significantly decreased in the PLS3E10-16del/0 rats, which was a fundamental structural determinant of bone strength. Second, higher porosity in cortical bone and disorganized collagen fibers in bone played important roles in impairment of bone strength(Gastaldi et al., 2020). (Page 9, Line 10-15)

The results were similar to previous studies:

WNT16-deficient mice displayed cortical bone defects with normal trabecular bone, with unchanged cortical MAR (Movérare-Skrtic et al., 2014). (Page 10, Line 13-15).

This study described that “Wnt16-deficient mice develop spontaneous fractures as a result of low cortical thickness and high cortical porosity.”[1]

References:

[1] Movérare-Skrtic, S., Henning, P., Liu, X., Nagano, K., Saito, H., Börjesson, A. E., . . . Ohlsson, C. (2014). Osteoblast-derived WNT16 represses osteoclastogenesis and prevents cortical bone fragility fractures. Nat Med, 20(11), 1279-1288. doi:10.1038/nm.3654